# STANDARDIZING THE MEASUREMENT OF TEXT DIVERSITY: A TOOL AND COMPARATIVE ANALYSIS

## ABSTRACT

The diversity across outputs generated by LLMs shapes perception of their quality and utility. Achieving high textual diversity in datasets is often a desired quality, but there is no standard method to measure this aspect of model behaviour. In this work we empirically investigate diversity scores on English texts and measure how much overlapping information is captured in these metrics. We find that computationally efficient compression algorithms capture information similar to what is measured by slow-to-compute $n$-gram overlap homogeneity scores. Further, a combination of measures—compression ratios, self-repetition of long $n$-grams and Self-BLEU and BERTScore—are sufficient to report, as they have low mutual correlation with each other. The applicability of scores extends beyond analysis of generative models; for example, we highlight applications on instruction-tuning datasets and human-produced texts. We release a diversity score package to facilitate research and invite consistency going forward.[1]

## 1 INTRODUCTION

Evaluation of LLM-generated texts is typically done with respect to accuracy or factuality, e.g., as measured via entailment (Tang et al., 2023), or text quality aspects such as coherence and fluency (e.g., estimated using LLMs as evaluators as in Liu et al. 2023). For summarization tasks where reference summaries are available, the similarity of generated outputs to these is also often measured. A complementary dimension of model performance is *diversity*, i.e., how much "boilerplate" content is repeated *across* LLM outputs.

The diversity of generated outputs is intuitively important: A model prone to repeating a few sentence constructions or boilerplate turns of phrase across its outputs will likely be deemed lower quality than an LLM that offers more diverse generations, all else being equal. For example, when prompted to summarize news articles, Llama-2 (Touvron et al., 2023b) tends to often generate text that begins with "The article discusses that...", "The article notes that...". Table 1 illustrates example repetitions from various models over a news summarization dataset.

This work primarily aims to standardize reporting of lexical diversity measures. We first analyze existing diversity scores over English language outputs from several LLMs. This identifies a few practical, (mostly) independent scores that characterize repetition. We highlight text length as an important factor when assessing diversity, and examine diversity in downstream datasets such as instruction tuning. We release an open-source Python package that can be used to explore and evaluate the diversity of generated text datasets.[1]

## 2 BACKGROUND

Lack of diversity may result from repetition of long stretches of text, or owe to subtle distributional patterns (Holtzman et al., 2019; Meister et al., 2022; 2023a). We focus on scores that are likely to capture overt repetition across outputs, and leave for future work similar analysis of semantic and structural diversity scores (Bär et al., 2012). Prior work on conditional generation tasks such as image captioning and dialog summarization have offered observations regarding the diversity of

---

[1] https://anonymous.4open.science/r/diversity-D36A/

| Model | Token Repetition Text | Pattern-Matched Text |
|---|---|---|
| StableLM | "The article also notes that..." `41/500`
"The article also mentions that..." `17/500`
"The article notes that the..." `15/500`
"The article concludes by stating..." `14/500`
"The author also notes that..." `11/500` | **DT NN VBZ DT JJ NN** `84/500`
"The article discusses the recent debate..."
"The book provides a helpful guide..."
"The article discusses the controversial penalty..."
"The article discusses the illicit market..." |
| FlanT5 | "The boy, who cannot be..." `5/500`
"The study, published in the..." `7/500`
"The body of a man who..." `3/500`
"A man has been charged with..." `5/500`
"... was last seen at a..." `4/500` | **NNP NNP DT JJ NN VBD** `37/500`
"Christopher Barry, a 53-year-old man, was..."
"Charles Collins, a 28-year-old man, saved..."
"Damian Parks, a 22-year-old student, went..."
"Lynn Fast, a 21-year-old mother, claimed..." |
| Llama-2 | "The article discusses the..." `15/500`
"The article also mentions that..." `28/500`
"The article concludes by noting..." `6/500`
"The article is about the..." `19/500`
"According to the article, a..." `16/500` | **NNP NN NNP NNP VBZ VBN** `42/500`
"American conductor Marin Alsop has become..."
"England striker Andy Cole has warned..."
"Barcelona defender Dani Alves has announced..."
"England economist Neil Haldane has said...." |

Table 1: Examples of exact text-match and repeating part-of-speech patterns in 500 model generated summaries of news articles from the CNN/DM dataset. The number of times the pattern occurs is shown in parenthesis. These patterns appear at a higher frequency in the generated outputs than in the original input data. Different models are characterized by a different set of repeated patterns.

produced texts. Prior work on both of these tasks has documented that models tend to repeat the same text for different contexts. Li et al. (2016) find that four phrases account for about a third of all turns produced by a conversational agent, and Devlin et al. (2015) report that more than half of automatically generated captions are repeated verbatim for different images. Self-repetition (Salkar et al., 2022), i.e., exact repetition of the same $n$-gram ($n \geq 4$) across different summaries is a practical way of measuring repetition for tasks where generated text is lengthier. In such cases exact matches rarely occur but repetition is common, especially relative to the training data (Wang et al., 2023a).

The above observations highlight a variety of metrics but it is unclear how to compare across varying definitions of lexical diversity. Compression ratios rely on repetition in strings to report how much a document can be reduced relative to the original size. In this work we propose a standardized approach to quantifying diversity, and show that *compression ratio*, a measure of document compression relative to its original size, is a fast, convenient to compute score that is sufficient to capture the information in all token/type ratio related alternatives. However, we also find that compression ratios—and all scores considered—are moderately to strongly correlated with the length of texts, complicating interpretation. Many comparisons remain meaningful when accompanied with information about length, but absent this variable no reliable conclusions can drawn.

The association between length and measures of diversity is well-established in corpus linguistics (for a detailed discussion, see Brysbaert et al. 2016). The number of unique words in a corpus is a power function of the total words seen, where the power is less than 1. The number of total words grows linearly, while the number of unique words is sublinear, so longer texts have more repetitions of unique words and $n$-grams than shorter texts (Covington & McFall, 2010; McCarthy & Jarvis, 2010).

LLM generations can reduce text diversity in a few ways. Guo et al. (2023) study the effect of consecutive rounds of distillation, in which a language model produces the data on which a separate language model is subsequently trained. They report dramatic reduction in diversity over 10 iterations. However, this study did not report the length of text produced in consecutive distillation rounds. Given our findings, it would be prudent to check if output lengths remain comparable in length.

Padmakumar & He (2023a) show that when people write with the aid of an LLM (e.g., instructGPT) they produce less diverse writing than when they do not. Here we ascertain that these results are independent of length, indicating that there is a genuine reduction in diversity (rather than merely affecting lengths which in turn influence diversity measures).

## 3 A SMORGASBORD OF TEXT DIVERSITY SCORES

The variety of scores used to measure diversity across a corpus of texts derive from two core ideas: Computing average similarity between pairs of outputs produced by the same model for different inputs, and computing variants of token/type ratio. The former are adapted from common approaches to text generation evaluation by comparing with references, using standard measures of pairwise similarity; the latter track the diversity of vocabulary measured as the ratio of unique words to total words produced, with the outputs from a model concatenated into a single text.

We first describe each score, and then present insights regarding their mutual redundancy. We also consider their required run-times, which are lengthy for some metrics and may render them impractical for analysis of a large number of outputs. All scores are defined for a set of generated texts $D$, each conditioned on its respective input.

**Self-BLEU**  The quality of text in machine translation, summarization, and image captioning is often reported in terms of overlap with a reference text. This idea can be adapted to measure diversity across different outputs by using one generated text as a reference and measuring the similarity of other outputs against this. Self-BLEU measures similarity between all text pairs in $D$ using BLEU as the similarity score (Zhu et al., 2018). BLEU can be replaced with an arbitrary similarity score, e.g., ROUGE-L or BERTScore. These variants are called **homogenization scores** and have recently been used to compare the diversity of texts produced under several conditions (Padmakumar & He, 2023a).

**Homogenization Score (ROUGE-L)**  All homogenization scores calculate an aggregate similarity across pairs of examples (Equation 1). Here the similarity score of choice is ROUGE-L Lin & Och 2004. This quantifies overlap in terms of longest common sub-sequences between all pairs of text in a corpus instead of the fixed $n$-gram size used in other ROUGE variants:

$$\text{hom}(D) = \frac{1}{|D| - 1} \sum_{d, d' \in D;\ d \neq d'} \text{sim}(d, d') \tag{1}$$

**Homogenization Score (BERTScore)**  This homogenization score uses BERTScore to measure similarity between documents in Equation 1. Unlike the other scores, it does not count the repetition of specific tokens, but instead uses BERT embeddings to (ideally) capture "semantic" similarity beyond verbatim $n$-gram matches.

**Self-repetition Score**  Self-repetition was introduced to measure the tendency of LMs to repeat long $n$-grams across different outputs (Salkar et al., 2022).

$$\text{SRS}(d) = \log \left( \sum_{i=1}^{k} N_i + 1 \right) \tag{2}$$

Where $k$ is total number of 4-grams in a single document $d \in D$, and $N_i$ the number of other summaries in which 4-gram $i$ appears. The final score is the sum of $\text{SRS}(d)$ divided by the number of documents in the corpus $D$.

**Moving Average Token-Type Ratio**  The token-type ratio for a text is the unique token count divided by the total count of tokens. Moving Average Token Type Ratios (MATTRs) measures the lexical dynamics across a text which is insensitive to text length. The score captures the repetition of a given word in segments of text and does not explicitly account for longer repeated sequences (Covington & McFall, 2010).

$N$**-Gram Diversity Score**  NGD extends the idea of token-type ratio to longer $n$-grams (Padmakumar & He, 2023a; Meister et al., 2023b; Li et al., 2023). It is defined as a ratio of unique $n$-gram counts to all $n$-gram counts:

$$\text{NGD}(D) = \sum_{n=1}^{4} \frac{\text{\# unique } n\text{-grams in } D\oplus}{\text{\# } n\text{-grams in } D\oplus} \tag{3}$$

Where $D\oplus$ denotes the dataset $D$ concatenated into a single string. We use four as the maximum $n$-gram length. This method captures repeated *sequences* in addition to single token diversity.

**Hypergeometric Distribution D**  The probability of text under a Hypergeometric Distribution D (HD-D) is an another measure of lexical diversity (McCarthy & Jarvis, 2010).[2]  HD-D does not capture repetition of sub-sequences.

**Compression Ratios**  The diversity scores introduced so far are all a function of the number of repeated substrings across outputs. We use gZip to compress the concatenated text of all outputs generated by a model. The compression ratio is then the ratio between the size of the compressed file to that of the original file. High compression ratios imply more redundancy:

$$\text{CR}(D) = \frac{\text{size of } D\oplus}{\text{compressed size of } D\oplus} \tag{4}$$

**Part-of-Speech Compression Ratio**  To capture repeated syntactic patterns, we also compute compression ratios for part-of-speech (POS) tag sequences. We use the NLTK POS tagger [3] and the Penn Treebank set of 36 tags.

## 4 DATA AND MODELS

We compute diversity scores for the outputs of six instruction tuned models on the CNN/DailyMail (Hermann et al., 2015) and XSUM (Narayan et al., 2018) English news summarization datasets. The models are: Llama-2 (Touvron et al., 2023a), GPT-4 (OpenAI, 2023), FlanT5-XXL (Longpre et al., 2023), StableLM (Taori et al., 2023; Chiang et al., 2023; Anand et al., 2023), Mistral (Jiang et al., 2023), and StableBeluga (Touvron et al., 2023b; Mukherjee et al., 2023). [4] We selected these models to cover a range of availability (open- and closed-source), and architectures (encoder-decoder, decoder-only).[5] The lengths of texts vary considerably by source, for reference and model-produced text alike, so we also note average lengths when reporting diversity.

## 5 TEXT LENGTH AS CONFOUNDER

To keep computational time and costs manageable, we randomly sample 500 inputs from CNN/DailyMail and XSUM for analysis. Table 2 reports diversity scores for the outputs generated by the six zero-shot LLMs for these inputs.

The top panel of Table 2 shows scores for human-written texts: the original article given as input for summarization, the baseline summary consisting of the first three sentences of the news article and the human reference summary. These scores serve as a reference point for the diversity scores of the models. One would expect that human-authored texts would be more diverse than those produced by LLMs (with the caveat that the texts were scraped from the web, and so may contain HTML, ads, and page layout artefacts that might be repetitive (Salkar et al., 2022)). The human texts differ by length and the sources of longer texts appear to be less diverse. The association between the length of the produced texts and their diversity is similarly pronounced in the XSUM dataset, as seen Table 3. Text length as a confounder for diversity has been reported in prior work (Salkar et al., 2022), along with potential methods to adjust for this, e.g., sampling blocks of fixed size (Covington & McFall, 2010).

---

[2]For both HD-D and MATTR, we use the implementation provided in the `lexical-diversity` package (`https://pypi.org/project/lexical-diversity/`).

[3]`https://www.nltk.org/api/nltk.tag.html`

[4]All models—except GPT-4—downloaded from HUGGINGFACE (`https://huggingface.co/models`).

[5]We use prompts for summarization provided by each model, where available. See Appendix A.3.

Figure 1: Correlations between text diversity scores on CNN/DM. Compression ratio correlates strongly with most other diversity metrics.

| Model | Avg. Length | CR (↓) | CR: POS (↓) | NGD (↑) | Self-Rep. (↓) | Hom. (R-L) (↓) | Hom. (BERT) (↓) | Self-BLEU (↓) | MATTR (↑) | HD-D (↑) |
|---|---|---|---|---|---|---|---|---|---|---|
| Article | 452.25 | 2.615 | 5.544 | 2.637 | 6.216 | 0.118 | 0.696 | 0.003 | 0.837 | 0.896 |
| Article (Lead 3) | 75.87 | 2.369 | 5.497 | 3.041 | 4.276 | 0.105 | 0.686 | 0 | 0.856 | 0.892 |
| Reference | 51.78 | 2.277 | 5.330 | 3.164 | 3.842 | 0.074 | 0.683 | 0 | 0.875 | 0.919 |
| StableLM | 132.71 | **2.724** | **5.940** | 2.673 | 4.940 | **0.126** | 0.689 | 0.002 | 0.792 | 0.867 |
| Mistral | 114.88 | 2.499 | **5.621** | 2.926 | 4.688 | **0.123** | _0.697_ | _0.036_ | 0.831 | 0.880 |
| Llama-2 | 106.52 | _2.543_ | **5.684** | _2.874_ | 4.159* | **0.125** | _0.694_ | 0.001 | _0.820_ | _0.873_ |
| StableBeluga | 91.17 | 2.452 | **5.644** | 3.028 | _4.467_ | **0.121** | _0.702_ | _0.047_ | 0.846 | 0.889 |
| FlanT5 | 63.84 | **2.453** | **5.608** | _2.939_ | 3.608* | 0.084 | 0.667 | 0 | _0.833_ | **0.887** |
| GPT-4 | 55.4 | 2.361 | **5.463** | 3.124 | _3.909_ | _0.098_ | _0.684_ | **_0.001_** | 0.853 | **0.891** |

Table 2: Diversity scores for the CNN/Daily Mail dataset. Arrows indicate direction of *more diversity*. Values indicating less diversity compared to at least one text source that produces longer human texts are bolded; models with scores that are less diverse than those from a model that produces longer summaries are underlined. An asterisk indicates a model more diverse than a shorter human text.

Table 5 reports correlations between the number of words produced by each model and diversity scores. All scores of the token/type ratio family are highly correlated with length, while the pairwise similarity ones are only moderately correlated. Self-BLEU has low correlation with length.

| Model | Avg. Length | CR (↓) | CR: POS (↓) | NGD (↑) | Self-Rep. (↓) | Hom. (R-L) (↓) | Hom. (BERT) (↓) | Self-BLEU (↓) | MATTR (↑) | HD-D (↑) |
|---|---|---|---|---|---|---|---|---|---|---|
| Article | 310.20 | 2.511 | 5.555 | 2.756 | 5.643 | 0.110 | 0.695 | 0.002 | 0.838 | 0.892 |
| Article (Lead-3) | 55.94 | 2.316 | 5.454 | 3.107 | 3.999 | 0.103 | 0.683 | 0 | 0.860 | 0.891 |
| Reference | 21.04 | 2.276 | 5.409 | 3.211 | 2.914 | 0.081 | 0.673 | 0 | 0.877 | 0.888 |
| StableLM | 109.20 | 2.745 | 6.008 | 2.636 | 4.687 | 0.130 | 0.695 | 0.002 | 0.78 | 0.854 |
| Llama-2 | 102.48 | 2.634 | 5.802 | 2.795 | 4.618 | 0.128 | 0.687 | 0.002 | 0.795 | 0.858 |
| Mistral | 95.18 | 2.531 | 5.708 | 2.911 | 4.495 | _0.132_ | _0.698_ | _0.044_ | 0.819 | 0.867 |
| StableBeluga | 88.46 | 2.461 | 5.673 | 2.992 | 4.418 | 0.124 | _0.698_ | _0.046_ | 0.837 | 0.88 |
| GPT-4 | 62.15 | 2.394 | 5.531* | 3.079 | 4.041 | 0.104 | 0.682 | 0 | 0.848 | 0.886 |
| FlanT5 | 20.93 | _**2.666**_ | _**6.222**_ | _**2.743**_ | _**2.868**_ | _**0.114**_ | 0.665 | 0.001 | _**0.756**_ | **0.842** |

Table 3: Diversity scores for XSUM summaries. Arrow indicate the direction of more diverse texts for each score.

| Model | CR (↓) | CR: POS (↓) | Self-Rep. (↓) | Hom. (BERT) (↓) | Self-BLEU (↓) |
|---|---|---|---|---|---|
| Article | 2.162 | 5.095 | 2.719 | 0.666 | 0 |
| Article (Lead 3) | 2.179 | 5.093 | 2.719 | 0.663 | 0 |
| Reference | 2.230 | 5.314 | 2.663 | 0.667 | 0 |
| Llama-2 | 2.345 | 5.636 | 2.919 | 0.663 | 0.002 |
| GPT-4 | 2.213 | 5.425 | 2.666 | 0.663 | 0 |
| FlanT5 | 2.490 | 5.737 | 2.707 | 0.665 | 0.001 |
| StableLM | 2.342 | 5.521 | 2.823 | 0.664 | 0.001 |
| Mistral | 2.308 | 5.689 | 2.736 | 0.659 | 0 |
| StableBeluga | 2.210 | 5.436 | 2.663 | 0.659 | 0 |

Table 4: Diversity metrics for XSUM summaries, with outputs from each model truncated to the length of the shortest. All scores are directly comparable.

| CR | CR: POS | NGD | Self-Rep. | Hom. (R-L) | Hom. (BERT) | Self-BLEU | MATTR | HD-D |
|---|---|---|---|---|---|---|---|---|
| 0.867 | 0.832 | 0.81 | 0.904 | 0.875 | 0.579 | 0.235 | 0.79 | 0.855 |

Table 5: Correlation between scores and total word counts (concatenated text) for CNN/Daily Mail.

## 6 DIVERSITY OF MODEL SUMMARIES

The confound of length complicates reporting. On both CNN/DM and XSUM (cf. Tables 2 and 3), StableLM produces the longest summaries. All scores indicate that these are the least diverse, most likely due to the length confound. In both sets of results, we look for models that produce shorter summaries that are less diverse. These findings are notable and hold, despite length differences.

Three types of differences are marked in the tables. Model summaries that are shorter but less diverse than human summaries are marked in bold. Human texts here are written by journalists, so the expectation is that they would be more diverse. More bold entries in a column indicate that the score captures differences between human and machine diversity, a desirable trait. Underlined entries denote models that are less diverse than other models that produce longer summaries. The more underlined entries there are for a model, the more indicators there are that its output is less diverse. Asterisks mark models that appear more diverse than a human text of shorter length.

The most interesting diversity scores are those that capture differences between human and automatically produced text, without necessarily committing to an interpretation of which source is preferable. Human evaluation in future work will address this question, which we discuss in 10. On the CNN/DM dataset, Hom. (BERT) and MATTR are the two scores that detect no differences between human and model texts. Compression ratio for part of speech sequences is the score that identifies the most differences between human and model-generated text. Self-repetition stands out as the only score that identifies model generated text as more diverse on the CNN/DM dataset. From this analysis, CR:POS and self-repetition emerge as prime candidates of reportable scores, while Hom. BERT is less useful.

## 7 CORRELATION ANALYSIS

We present three sets of correlation analyses between *(i)* different diversity scores, *(ii)* the same diversity score across datasets, and *(iii)* diversity scores and standard reference-based evaluations. Despite the large number of diversity scores in our list, they all revolve around $n$-gram repetition. It is of interest to know if they capture different information. With this motivation, we compute the correlations between every pair of scores, reported in Figure 1.

Compression ratio is highly to moderately correlated with other $n$-gram scores. The only weak correlations are with Self-BLEU and Hom. (BERT). Given the degenerate behavior of Hom. (BERT) on the analysis of summaries, reporting Self-BLEU only is advisable. Finally, self-repetition is only moderately correlated with with other scores, so is informative to report as a standard score for diversity. The correlations are similar on the XSUM summaries (see Appendix 4), reinforcing the recommendation for the set of scores that should be used to capture diversity. Diversity analysis on

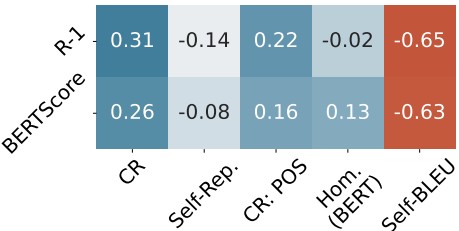

Figure 2: Correlations between diversity metrics, BERTScore, and ROUGE-1. Both reference-based metrics are weakly correlated with CR and Hom. (BERT), and moderately anti-correlated with Self-BLEU.

| Model | CR (↓) | CR: POS (↓) | Self-Rep. (↓) | Hom. (BERT) (↓) | Self-BLEU (↓) |
|---|---|---|---|---|---|
| Article | 2.268 | 5.25 | 2.763 | 0.676 | 0 |
| Article (Lead 3) | 2.274 | 5.25 | 2.762 | 0.658 | 0 |
| Reference | 2.189 | 5.179 | 2.763 | 0.674 | 0 |
| Llama-2 | 2.96 | 5.627 | 2.847 | 0.674 | 0.001 |
| GPT-4 | **2.287** | 5.376 | 2.761 | 0.672 | **0** |
| FlanT5 | 2.288 | 5.389 | 2.779 | 0.673 | **0** |
| StableLM | 2.393 | 5.537 | 2.884 | 0.672 | 0.001 |
| Mistral | 2.32 | 5.415 | 2.812 | **0.67** | **0** |
| StableBeluga | 2.288 | 5.46 | 2.766 | 0.671 | **0** |

Table 6: Scores on CNN/DM summaries truncated to the shortest summary length for a given input.

the CNN/DM and XSUM datasets did not indicate consistent system behavior. We report the analysis in Table 10.

Our guiding assumption is that output diversity and self-repetition are aspects of model behavior that are not captured by existing evaluation approaches. Here we directly test this assumption. We compute the system level correlation between the diversity scores and the traditional BERTScore and ROUGE evals, shown in Figure 2. Reference-based evaluations are only weakly correlated with the diversity metrics. Self-BLEU, however, is moderately anti-correlated with with both ROUGE-1 and BERTScore.

# 8 TRUNCATING TO CONTROL LENGTH

For each input for summarization, we truncate all summaries to the length of the shortest one produced by any of the sources as a crude means to remove the influence of length on scores. The resulting scores are directly comparable across sources, listed in Tables 4 and 4. Compression ratio and Self-BLEU scores indicate that model-produced text is less diverse than human text. Hom. (BERT) scores barely vary across sources, further supporting the recommendation that this is not a useful score to report. On the CNN/DM dataset, Self-BLEU indicates that Llama-2 and StableLM are the most repetitive models. Compression ratio also ranks these two models as the least diverse. The results are consistent on XSUM, but for that dataset Flan-T5 is also highly ranked and the most repetitive.

The truncation approach to control for length is not practical for published research or leaderboards. Introducing a new source of texts would require recomputing the scores for other sources one may want to compare with, which is impractical and sometimes impossible when the outputs from other sources are not available. Future research will have to search for more practical alternatives.

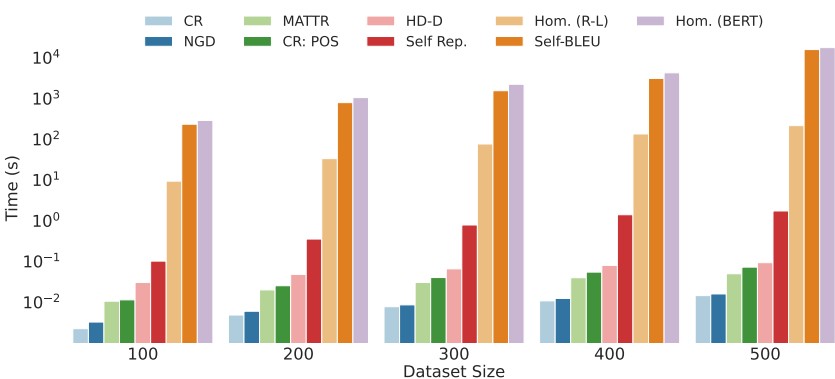

Figure 3: Mean run time **(log-scale)** on CNN/DM summaries. Run times increase with the number of text for the analysis. Even for small datasets, Self-BLEU and BERTScore homogenisation are slow.

## 9 RUN-TIME CONSIDERATIONS

When analyzing the diversity of large volumes of text, run-time considerations become relevant. Figure 3 provides insights about the feasibility of obtaining scores for large samples[6]. The compression ratio scores are fast, with text compression utilities specifically optimized for speed. Self-repetition takes longer but acceptable time. Self-BLEU and Hom. (BERT) are prohibitively slow.

## 10 BROADER APPLICATIONS

The guiding motivation for this work has been to develop standardized and informed approach to the analysis the diversity of text produced by LLMs. The standardization of scores will facilitate analysis in broader settings. Here we provide two examples: human writing, with and without facilitation from a LLM, and instruction tuning datasets.

**Human Story Writing**    Padmakumar et al. (2023) presented an analysis of human-written stories, where people wrote either by themselves or with the help of GPT-3 or GPT-3.5 Turbo. They find that using LLMs as writing partners leads to greater homogenization of the stories. As reported by Padmakumar et al. (2023), we find that all diversity scores agree that people writing independently produce the more diverse texts (cf. Table 7). Here, story length is not an issue because the average length of stories in each setting are comparable: 375 words for writing without help, 372 words when writing with GPT-3 and 370 when writing with GPT-3.5.

| Dataset | CR (↓) | CR: POS (↓) | Self-Rep. (↓) | Hom. (BERT) (↓) | Self-BLEU (↓) |
|---|---|---|---|---|---|
| Solo | 2.901 | 5.314 | 5.873 | 0.604 | 0.018 |
| GPT-3 | 2.940 | 5.371 | 5.911 | 0.613 | 0.020 |
| InstructGPT | 3.064 | 5.462 | 5.966 | 0.631 | 0.022 |

Table 7: Diversity scores over essays. Working with an LLM correlates with lower diversity.

**Instruction-tuning Datasets**    The quality and diversity of instructions are likely to result in more robust and capable systems (Sanh et al., 2022; Mishra et al., 2022). We analyze the diversity of five instruction-tuning datasets: Open Assistant (Köpf et al., 2024), Super-NaturalInstructions (Wang et al., 2022), Unnatural Instructions (Honovich et al., 2023), Alpaca (Wang et al., 2023b), and Dolly (Conover et al., 2023). [7].

In Table 8 we report diversity scores. Here datasets are ordered by size; we therefore expect that scores will be sorted in diminishing order in each column. Only deviations from this ordering are

---

[6]Run on a single NVIDIA Quadro RTX 8000 GPU.

[7]We provide further details in Appendix A.5

| Dataset | CR ($\downarrow$) | CR: POS ($\downarrow$) | Self-Rep. ($\downarrow$) |
|---|---|---|---|
| Open Assistant | 2.886 | 6.731 | 3.969 |
| Unnatural Instructions | 4.191 | 7.278 | 9.868 |
| Alpaca | 3.119 | 6.61 | 3.105 |
| Super-NaturalInstructions | 2.675 | 5.749 | 3.456 |
| Dolly | 2.578 | 6.214 | 2.935 |

Table 8: Diversity scores for instruction datasets. We do not include Self-BLEU nor Hom. (BERT) due to long run times. Datasets are ordered by size and differ vastly in length, so only scores for which a smaller dataset is less diverse are meaningfully interpretable.

reportable. We provide details about the number of instructions and words in Appendix A.5. Open Assistant instructions are remarkably diverse compared to the other datasets, and all diversity scores for it are more favorable than that for other datasets. Unnatural instructions are remarkable in the opposite direction, with outlier scores that are so much higher, they are likely not due to length entirely. We provide an analysis of the diversity scores with the length controlled in Appendix A.6.

Given the large dataset sizes, ranging from 15-80k data points, we do not compute the homogenization scores nor Self-BLEU, as the computation time is infeasible. For approximately 50k instructions, the estimated computation times ranged from 48 to 800 hours for these scores. This case study highlights the relevancy of the run-time analysis for computing score that we presented in the previous section.

**Human Assessments of Diversity**    Human judgements of diversity are difficult to reliably collect. Prior work shows that humans tend to implicitly conflate quality of text with its diversity, and it can be difficult to separate content and lexical diversity in these assessments (Tevet & Berant, 2021).

This work primarily aims to standardize reporting of lexical diversity metrics, and in doing so, measure the overlap in the information that is captured by each. Doing so ensures that validity of the various reported lexical metrics, which we believe should necessarily come before measuring correlation to human judgements of diversity.

## 11    DISCUSSION AND RECOMMENDATIONS

Our in-depth analyses reveal that compression ratio is an excellent score to report, easy to compute and strongly correlated with other scores used in past work. Compression ratio of part of speech sequences capture differences between human and model-generated text. Self-repetition zeros in only on repetition of longer $n$-grams across generations, and is only moderately correlated with compression ratios. Finally Self-BLEU is only weakly correlated with the previous three, so is a good complement score to report. In our analyses, we identified several drawbacks of BERTScore: it does not show differences between human and model-generated text and barely varies when adjusted for length. There is no good justification to report it.

Length of the analyzed text has to be reported alongside all these scores. When length differs, scores are not meaningfully comparable. Truncating and downsampling text is one way to produce a set of results that are intuitively comparable. Different random draws of the sample chosen to represent a dataset will likely differ in diversity; the selection may lead to unwarranted conclusions. Truncating texts prevents any possibility of discovering repetitive behavior towards the end of longer text. Future research into a principled solution for this problem is urgently needed. Despite all this, we were able to glean meaningful insights about differences in diversity between human and model-produced text for summaries, essays and instructions.

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

# A   APPENDIX

## A.1   EXAMPLES OF REPETITIVE PATTERNS

Table 9 show more examples of repeated sentence structures (using part-of-speech tags) from Padmakumar et al. (2023).

| Dataset | Token Repetition Text | Pattern-Matched Text |
|---|---|---|
| GPT-3 | "In my opinion..." 41/100 | **PRP VBZ RB JJ TO VB** 15/100
"It is also vital to discern..."
"It is very easy to construct..."
"It is largely inappropriate to try..."
"It is morally acceptable to focus..."
**PRP VBP IN DT NN IN** 12/100
"I don't like the damsel in..."
"I fear that a cycle of..."
"I feel that an acknowledgement of..."
"I find that the inflection of..." |
| Instruct-GPT | "In my opinion..." 25/100
"It is important to..." 20/100
"Up with the news..." 15/100 | **MD VB DT JJ NN IN** 20/100
"...can have a huge variety of..."
"...can have a negative effect on..."
"...can have a positive impact on..."
"...can have a sturdy framework for..."
**PRP VBZ RB JJ TO VB** 12/100
"It is also important to realize..."
"It is fairly common to hear..."
"It is indeed surprising to hear..."
"It is probably impossible to keep..."
"...it becomes very cringy to watch..." |
| Solo | "In my opinion..." 22/100
"In my opinion, the..." 13/100
"When it comes to..." 11/100
"In my opinion, I..." 10/100 | **PRP VBZ JJ TO VB IN** 9/100
"It is crucial to recognize that..."
"It is crucial to remember that..."
"It is unjustifiable to assume that..."
"It is important to acknowledge that..."
**PRP VBP IN DT JJ NN** 10/100
"I believe for the right person..."
"I do on a regular basis."
"I fall into the second group."
"I live in a small town..." |

Table 9: Examples of exact text-match and repeating part-of-speech patterns in essays from Padmakumar & He (2023b). The number of times the pattern occurs is shown in parenthesis.

## A.2   CORRELATION BETWEEN METRICS

Self-BLEU scores are almost perfectly correlated between the two datasets; they appear to not be affected by text source. The other scores are still moderately to highly correlated but as already observed, models are ranked differently. When reporting diversity, source of analyzed data also has to be taken into account, in addition to length.

| CR | CR: POS | NGD | Self-Rep. | Hom. (R-L) | Hom. (BERT) | Self-BLEU | MATTR | HD-D |
|---|---|---|---|---|---|---|---|---|
| 0.83 | 0.695 | 0.885 | 0.87 | 0.841 | 0.921 | 0.991 | 0.799 | 0.654 |

Table 10: Score correlations for each text diversity score between the CNN/DM and XSUM datasets.

### A.3 SUMMARIZATION PROMPTS

Table 11 details the prompts and format used to generate the summaries for the news datasets. We follow the formats recommended provided by each model, and insert the along with the instruction for summarization.

| Model | Model Size | Prompt |
|---|---|---|
| Llama-2 | 7B | `[TEXT]` [INST] Summarize the above text. [/INST] |
| GPT-4 | - | `[TEXT]`. Summarize the above text. |
| Flan-T5 | 11B | Summarize this article: `[TEXT]` |
| StableLM | 7B | `[TEXT]` < \|USER\| >Summarize the above text. < \|ASSISTANT\| > |
| Mistral | 7B | ### Instruction: Summarize the following: ### Input: `[TEXT]`. ### Response: |

Table 11: Prompts used for each model to generate a summary. `[TEXT]` is replaced with the input article.

### A.4 XSUM METRICS

Figure 4 shows the correlations between all pairs of metrics for the XSUM dataset. The correlations show that compression ratio is highly to moderately correlated with other n-gram scores, similar to the findings for the CNN/DM dataset.

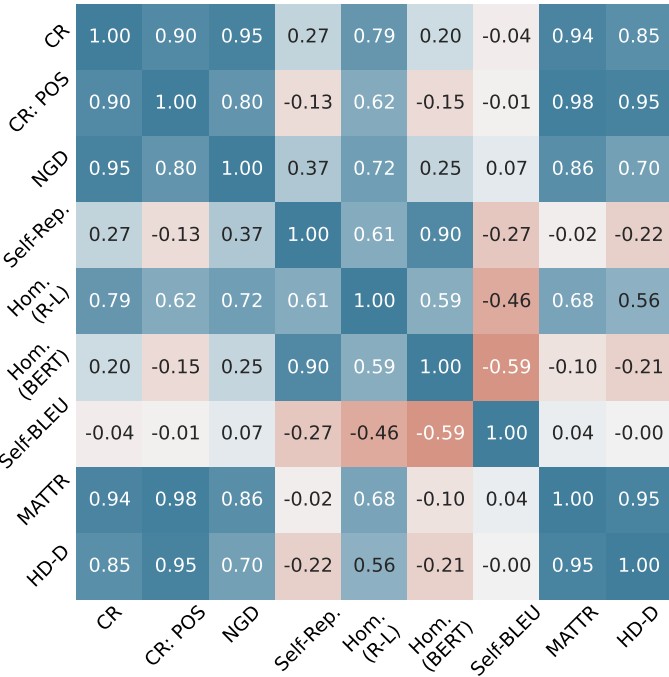

Figure 4: Correlation table between scores on XSUM.

### A.5 INSTRUCTION DATASETS, DETAILS

**Open Assistant** is a collection of crowdsourced instructions (Köpf et al., 2024). The data was collected under detailed guidelines and includes questions that reflect real-life situations.

**Super-NaturalInstructions**   A corpus comprising crowdsourced instructions that transform 200 benchmarks and intermediate evaluation results into a set of instructions and demonstrations (Wang et al., 2022).

**Unnatural Instructions**   An (almost) automatically created dataset, using instructions from the SuperNatural-Instructions dataset to automatically generate new instructions (Honovich et al., 2023). To increase diversity, each instruction was also paraphrased. Honovich et al. (2023) compare the diversity of instructions in Unnatural and Super-Natural Instructions with pairwise BERTScore similarities (within each dataset), and find that the similarities are much higher in Super-NaturalInstructions.

**Alpaca**   This dataset is created following the Self-instruct dataset (Wang et al., 2023b). GPT-3 was prompted to create instructions and demonstrations based on a seed of 175 human-written instructions. Crucially, the collection method includes a diversity filter, only including model-written instructions if their ROUGE-L similarity is less than 0.7 with an existing instruction. Length of instructions and demonstrations is also controlled for as a criterion for inclusion in the final instruction dataset.

**Dolly**   A set of human instructions and demonstrations, collected by Datrabricks employees (Conover et al., 2023). By design, they cover only eight classes of popular tasks: creative writing, closed and open QA, summarization, information extraction, classification and brainstorming.

Table 12 shows the number of instructions, the typical length of an instruction and average number of words per instruction set. All vary, making it even harder to control for length. Truncating makes less sense here, and down-sampling the number per instructions is counter-productive given our goal to understand the diversity of the entire dataset. We do make use of these instruments given the lack of alternatives, but note that more meaningful solutions are urgently needed.

| Dataset | # Instructions | Avg. # Words | Total # Words |
|---|---|---|---|
| Open Assistant | 84,437 | 78.10 | 6,594,646 |
| Unnatural Instructions | 66,010 | 38.05 | 2,511,737 |
| Alpaca | 52,002 | 10.06 | 523,329 |
| Super-NaturalInstructions | 4550 | 92.58 | 421,228 |
| Dolly | 15,011 | 12.37 | 185,816 |

Table 12: Average number of words, and size of the instruction datasets. Numbers correspond to the training set available from Huggingface. For Super-NaturalInstructions, we filter for English-only instructions using the `langdetect` library.

### A.6   INSTRUCTION DATASETS, LENGTH CONTROLLED

Table 13 shows scores for instructions downsampled to the size of the smallest dataset, and truncated to the length of the shortest instructions in the remaining data. Again, the Open Assistant dataset stand out as most diverse, while the Unnatural Instructions dataset is markedly less diverse than the others. Self-repetition in the related Super-Natural and Unnatural instructions is notably high. The human instructions in Dolly compare favorably with automatic instructions, especially when bearing in mind that only eight tasks are covered in it. CR:POS points to Super-natural instructions as the most diverse. We do not have a convincing explanation of why it compares so favorably against others on this score.

| Dataset | CR ($\downarrow$) | CR: POS ($\downarrow$) | Self-Rep. ($\downarrow$) |
|---|---|---|---|
| Open Assistant | 2.370 | 5.402 | 1.741 |
| Unnatural Instructions | 6.036 | 8.421 | 5.595 |
| Alpaca | 3.301 | 6.044 | 2.020 |
| Super-NaturalInstructions | 2.458 | 1.844 | 4.859 |
| Dolly | 2.832 | 5.504 | 2.235 |

Table 13: Truncated diversity scores for instruction datasets.

