# OpenReview forum: "Standardizing the Measurement of Text Diversity: A Tool and Comparative Analysis"
_ICLR.cc/2025/Conference — ICLR 2025 Conference Withdrawn Submission_

### Official Review · Reviewer_au29 · 2024-10-31

**Soundness:** 3
**Presentation:** 2
**Contribution:** 2
**Rating:** 5
**Confidence:** 3

**Summary:**

The paper explores an interesting perspective of model evaluation, focusing on text diversity as opposed to traditional metrics like accuracy or factuality. It analyses various text diversity metrics, including the repetition of n-gram patterns and its variant like POS tag sequences, and the uniqueness ratio of patterns across documents. The compression rate is also analysed, treating documents as a whole. Regarding the complex and vague definition of “diversity”, the study is founded on the assumption that human-generated text is more diverse than text generated by models. It evaluates both types on text summarization tasks using the above mentioned benchmarks.

**Strengths:**

- This study focus on an interesting and under-explored area in text evaluation.
- The paper presents an empirical study and analysis of different text diversity metrics. It finds that metrics like compression ratio and POS sequence can reveal difference between human and model-generated text. Metrics such as self-repetition and Self-BLEU, less correlated with others, offer valuable complementary insights. BERTScore shows minimal differentiation between human and model generated text, and also less in variation with text length, suggesting it may not be a worthwhile metric in this context.

**Weaknesses:**

- Although the paper focus on an interesting topic, the research is somewhat limited regarding providing minimal guidance for improving training or decoding strategies. The authors themselves also acknowledge a lack of comprehensive comparison of length-variant metrics.
- The introduction of metrics in Section 3 could be improved. The authors could consider including mathematical equations for each metric and provide a unified view.

**Questions:**

Section 10 provides interesting insights related to practical applications. Have the authors considered sampling an equal amount of instruction-response data from different instruction datasets, ensuring the same lengths, to compare diversity? Although there are challenges like controlling sentence length exactly, a comparison in relatively fair circumstances would be valuable.

---

### Official Review · Reviewer_wSbM · 2024-11-05

**Soundness:** 2
**Presentation:** 3
**Contribution:** 2
**Rating:** 3
**Confidence:** 4

**Summary:**

This work presents an empirical study on different strategies of measuring text diversity, and argues that compression ratio can be a good measurement and a proxy to other metrics.

**Strengths:**

- This paper investigates a collection of diversity metrics, and comparing them across the same setup is interesting.
- The empirical results on several benchmark datasets are informative.

**Weaknesses:**

- The claim of compression ratio being the final evaluation metrics is less convincing. For example, the correlation coefficients are not significant enough with some metrics, such as Self-BLEU and Hom (BERT).
- Furthermore, a more theoretical foundation needs to be provided to justify the which one being the a good proxy to all the rest and why, as I am afraid the correlation study may also be data dependent.
- The lack of human evaluation makes the foundation of this work less solid. After all, these are all automatic evaluation metrics. I noticed the discussion in section 10 about human assessments of diversity, which I believe the same observations exist in any other types of human evaluation on text generation. However, I think the challenge, to some extent, can be addressed by experiment design. Regardless, some results from human evaluation would be great.

**Questions:**

Please refer to the weaknesses section.

---

### Official Review · Reviewer_ZTJr · 2024-11-09

**Soundness:** 3
**Presentation:** 2
**Contribution:** 2
**Rating:** 3
**Confidence:** 4

**Summary:**

The authors focus on the problem of computing the textual diversity of responses generated by LLMs. They evaluate various textual diversity measures that rely on two core ideas: a. Computing the overlap between the different outputs of the same LLM, b. Computing the token-to-type ratio. They evaluate the diversity scores of LLMs for the task of summarization using two datasets, six LLMs covering both open-source and closed-source models. Their findings suggest that compression ratio based scores which are faster to compute and correlated with other diversity scores are more suitable for measuring diversity in the responses of LLMs.

**Strengths:**

A thorough study of existing diversity scores.

**Weaknesses:**

1. The paper lacks novelty. It is completely acceptable for a paper not to introduce a new method or metric if it performs a detailed analysis of existing metrics. However, in that case, I would expect the results to be novel or surprising, which is not the case here. For example, the finding that length is a confounder is already known. Further, in the absence of comparing with human correlations, it is not clear if the paper can clearly recommend one or more diversity scores which are most suitable.
2. The overall contribution does not seem substantial for an ICLR paper.

**Questions:**

1. The experiment on truncating the length does not seem sound. Arbitrarily truncating length to match the shortest response does not make sense. It would lead to incoherent and incorrect summaries, which would not be meaningful for any further analysis.
2. It is mentioned that the OpenAssist dataset has the highest diversity in prompts, but the overall scores seem to be the best for Dolly. Am I missing something?
3. As the authors rightly mention, the validity or utility of a diversity score should be judged based on its correlation with human judgments. However, no such study is done (and I understand the difficulty of doing so). In the absence of such a study, it is difficult to pick one or more metrics that would be most suitable for this task.
4. There are better metrics than BERTScore and R1 that can be used for evaluating summarization. The authors could consider these metrics for their analysis.
5. Suggestion: The tables and figures are often not very close to the text where they are being referred to. Some reorganization may improve readability.

---

### Official Review · Reviewer_7T4m · 2024-11-09

**Soundness:** 2
**Presentation:** 3
**Contribution:** 2
**Rating:** 3
**Confidence:** 4

**Summary:**

This paper aims to characterize textual diversity metrics in order to standardize reporting of diversity metrics as a complement to traditional, commonly used metrics for quality. They investigate the quality of and correlations between several types of diversity metrics, from a class of metrics called homogenization scores to compression ratios which are defined with respect to compression algorithms such as gZip. They find that many of these metrics have low correlations with each other, and recommend reporting a subset of them. Moreover, they find that length is a strong confounding factor when calculating diversity scores and provide some guidelines on how to control for it when calculating diversity scores.

**Strengths:**

1. A comprehensive set of diversity metrics are compared, and the run-time analysis is a helpful contribution to the literature on this topic.
2. The authors provide reasonable, concrete recommendations for practitioners in terms of which diversity scores are most helpful to report, after taking mutual redundancy between the metrics into account.

**Weaknesses:**

1. There is a lack of detail regarding experimental hyper-parameters that are very important and relevant to the investigation, such as decoding method and hyper-parameters relating to decoding (e.g. temperature or top_p/top_k parameters, if used)
2. Comparison against human-written texts is prevalent throughout the paper, but the comparisons seem unfair. For example, many recent works (e.g. Goyal et al. (2022) and Zhang et al. (2024)) demonstrate the flaws in the "human-written" reference summaries in CNN/DailyMail and XSum. Most importantly, they re-iterate that the references in these datasets were not necessarily actual summaries. Thus, there is reason to suspect the resulting diversity scores calculated across these human references is inflated.
3. While the correlation analysis is helpful, the paper is missing a strong validation component. Due to concerns such as the one listed above, I do not think comparison against references provided in CNN/DM and XSum provide this. Even if the authors consider further comparison against human-written text to be out-of-scope for this paper, I think some kind of human validation would be useful. For example, which metrics correlate the best with human judgments of diversity? This would not involve collecting more human-written text, but rather just judgments of diversity. Alternatively, the authors could examine the quality of these diversity metrics by manipulating parameters such as temperature (which necessarily increase/decrease diversity) and then examining whether the metrics reliably reflect the resulting changes in diversity.
4. I am not sure whether truncation is the best way for controlling for length. For instance, the authors could also investigate adding length constraints to the prompt given to each model as well.
5. Some citations to recent work that are very relevant are missing, such as:
"What Comes Next? Evaluating Uncertainty in Neural Text Generators Against Human Production Variability" (Giulianelli et al., 2023)
"Compare without Despair: Reliable Preference Evaluation with Generation Separability" (Ghosh et al., 2024)
"The Vendi Score: A Diversity Evaluation Metric for Machine Learning" (Friedman and Dieng, 2023)
6. It would be useful to see experiments for other tasks in the main body of the paper. I am aware that the authors mention that their focus is not on metrics that measure semantic/structural diversity, but in light of that, I am not sure if summarization is the best task for this investigation. This is because it seems natural that summaries will contain repeated text more often (e.g. "This article says..."). While I understand that this is not necessarily desirable, I think it is more important if generations for other, more open-ended, tasks contain repeated n-grams.

**Questions:**

1. Just to clarify, diversity is calculated with respect to outputs over several instances? Was only 1 output sampled for each input?
2. What were the decoding settings used for the summarization experiments?
3. BLEU is sensitive to which generation is chosen as the reference and which one is chosen as the hypothesis. For Self-BLEU, where such a distinction does not naturally exist, how did you determine which generation was chosen as the reference/hypothesis?

---

### Official Review · Reviewer_8XWv · 2024-11-13

**Soundness:** 2
**Presentation:** 3
**Contribution:** 2
**Rating:** 3
**Confidence:** 4

**Summary:**

This paper focuses on the diversity metric of generated texts and empirically investigates several existing diversity scores. The authors find that compression algorithms capture information similar to what is measured by slow-to-compute n-gram overlap homogeneity scores. Extensive experiments are conducted on instruction-tuning datasets and human-produced texts to provide some insights into the evaluation of diversity.

**Strengths:**

1. The proposed diversity metric based on compression ratio of gZip is interesting and worth further exploring.

**Weaknesses:**

1. This paper seems like an empirical try rather than standardization. First, the intuition of using compression ratios is vague. The authors directly adopt compression ratios for evaluating diversity of generated texts without delving into the principle of gZip compression, which makes this method less supportive. Then, the authors compare different diversity scores and show some relationships (such as correlations) among them. But in my view, the authors fail to provide a standard way to measure diversity of texts in terms of the advantages and disadvantages of different metrics.

2. This paper mentions the length of generated texts for many times to highlight its important role in the evaluation of diversity. However, this paper lacks discussion on the related works which focus on alleviating the length bias when measuring diversity, such as [1]. The authors should definitely incorporate this line of works into their paper.

3. The experimental part should be further improved. The current experimental results neither demonstrate the effectiveness of the metric based on compression ratios compared with other diversity metrics, nor show the complementation of all these diversity metrics. Since the position of this paper is “Standardizing the Measurement of Text Diversity”, the authors should at least standardize an experimental setting to compare the performance of different diversity metrics fairly.

[1] Rethinking and Refining the Distinct Metric. ACL 2022.

**Questions:**

My questions have been included in the weaknesses part.

---

### Note · Authors · 2024-11-20

**Comment:**

Revise and resubmit

**Withdrawal Confirmation:**

I have read and agree with the venue's withdrawal policy on behalf of myself and my co-authors.